# Comparison of Rhizosphere Bacterial Communities of *Pinus squamata*, a Plant Species with Extremely Small Populations (PSESP) in Different Conservation Sites

**DOI:** 10.3390/microorganisms12040638

**Published:** 2024-03-22

**Authors:** Fengrong Li, Shugang Lu, Weibang Sun

**Affiliations:** 1School of Life Sciences, Yunnan University, Kunming 650091, China; lifengrong@mail.kib.ac.cn; 2Yunnan Key Laboratory for Integrative Conservation of Plant Species with Extremely Small Populations, Kunming Institute of Botany, Chinese Academy of Sciences, Kunming 650201, China; 3Key Laboratory for Plant Diversity and Biogeography of East Asia, Kunming Institute of Botany, Chinese Academy of Sciences, Kunming 650201, China; 4University of Chinese Academy of Sciences, Beijing 100049, China

**Keywords:** *Pinus squamata*, rhizosphere, bacterial community, diversity, different conservation sites

## Abstract

*Pinus squamata* is one of the most threatened conifer species in the world. It is endemic to northeastern Yunnan Province, China, and has been prioritized as a Plant Species with Extremely Small Populations (PSESP). The integrated study of soil properties and rhizosphere bacteria can assist conservation to understand the required conditions for the protection and survival of rare and endangered species. However, differences between the rhizospheric bacterial communities found in the soil surrounding *P. squamata* at different conservation sites remain unclear. In this study, Samples were collected from wild, ex situ, and reintroduced sites. Bacterial communities in different conservation sites of *P. squamata* rhizosphere soils were compared using Illumina sequencing. The soil physicochemical properties were determined, the relationships between the bacterial communities and soil physicochemical factors were analyzed, and the potential bacterial ecological functions were predicted. The reintroduced site Qiaojia (RQ) had the highest richness and diversity of bacterial community. Actinobacteria, Proteobacteria, and Acidobacteriota were the dominant phyla, and *Bradyrhizobium*, *Mycobacterium*, *Acidothermus* were the most abundant genera. Samples were scattered (R = 0.93, *p* = 0.001), indicating significant difference between the different conservation sites. The abundance of *Mycobacterium* differed between sites (0.01 < *p* ≤ 0.05), and the relative abundances of *Bradyrhizobium* and *Acidothermus* differed significantly among different sites (0.001 < *p* ≤ 0.01). Soil total potassium (TK) and available nitrogen (AN) were the main factors driving bacterial community at the phylum level (0.01 < *p* ≤ 0.05). This study generated the first insights into the diversity, compositions, and potential functions of bacterial communities associated with the rhizosphere soils of *P. squamata* in different conservation sites and provides a foundation to assess the effect of conservation based on bacterial diversity and plant growth-promoting rhizobacteria (PGPR) to guide future research into the conservation of *P. squamata*.

## 1. Introduction

China is the center of gymnosperms diversity [1], and the Chinese flora includes many ancient relict species, including *Ginkgo biloba*, *Metasequoia glyptostroboides*, *Glyptostrobus pensilis*, *Cathaya argyrophylla*, *Pseudolarix amabilis* and others. However, many species are severely threatened; the top three threats to endangered gymnosperm species are habitat degradation, restricted distribution and overexploitation [2]. *Pinus* Sect. *Strobus* is thought to have evolved in China [3]. The section is distinguished by the needles, which are in clusters of five, and by the single-vascular bundle. Sect. *Strobus* is therefore considered to be the earliest-diverged and most ancient lineage of the Pinaceae. The threats facing the species of *Pinus* Sect. *Strobus* and the circumstances underlying their scarcity have been studied [4].

*Pinus squamata*, also known as the Qiaojia five-needle pine, was published as a new species in 1992 [5]. The tree has mottled bark and a straight trunk, and has high ornamental and economic value. Currently, only 34 individuals of *P. squamata* are known from the wild. The species has been assessed as Critically Endangered (CR) according to the endangered species classification system in 2001. Indeed, at the 24th World Congress on Conservation, the IUCN included *P. squamata* on the list of the 100 most endangered species in the world [6].

PSESP has a small remaining population, a restricted narrow habitat, severe anthropogenic disturbance and is at a high risk of extinction [7,8]. It should be emphasized that naturally rare species do not qualify as PSESPs, and external disturbance is necessary to determine whether a species is included as a PSESP [9]. There have been a number of programs and projects implemented at a national and provincial level to rescue and conserve PSESPs [10,11], and conservation biologists have been paying close attention to PSESP and conservation action plans in China [12,13,14]. As one of 20 key species of PSESPs in Yunnan, *P. squamata* was in 2011 included on the national list of 120 PSESPs in China [7,15].

The rhizosphere, which is the soil directly surrounding plant roots, is an important zone for the interactions between plants, soil and microorganisms [16]. Rhizosphere soils provide natural microhabitats for diverse microorganisms and are therefore considered to be hotspots of microbial diversity and activity in soils [17]. The rhizosphere microbiome affects plant growth and development, as well as resistances to stress and efficiency of nutrient use, therefore playing an important role in promoting soil quality and plant health [18,19]. Rhizobacteria that promote plant growth and health are known as plant growth-promoting rhizobacteria (PGPR) [20]. They have a crucial role in soil quality and fertility, as well as in the management of abiotic and biotic stresses [21]. Plants are directly affected by PGPR through the increase in nutrient availability, but they may also be indirectly affected by PGPR competing with pathogens [22]. PGPR are under consideration as biostimulants for sustainable agriculture, alleviators of abiotic stresses in soil, and green bioinoculants and biofertilizers [23,24,25].

With the drastic changes occurring in the environment, such as drought, soil biodiversity is variational in a rapidly changing world [26]. Adverse subsurface conditions may lead to phenotypic decline in species, which can be a cause of endangerment and extinction. Therefore, the integrated study of soil properties and rhizosphere bacteria can assist conservation, as it allows us to understand the conditions required for conservation of rare and endangered species. A comparative analysis of rhizospheric fungi between wild, ex situ, and reintroduced *P. squamata* was researched by Li and Sun [27]. To date, little is known about the rhizospheric bacterial communities found in the soil of *P. squamata*.

We hypothesized that the rhizosphere bacteria will have different effects on the wild, ex situ, and reintroduced individuals of *P. squamata*, and that the endangered cause of *P. squamata* may be lower bacterial diversity, fewer beneficial taxa, and low soil nutrients in the rhizosphere soils. Different conservation sites of *P. squamata* vary widely in bacterial composition, so rhizosphere bacteria can be used to assess the effect of conservation and guide conservation practices. It is our hope that this study will inform the further optimization of conservation approaches and strategies, and that relevant plant growth-promoting bacteria (PGPR) will guide research into managing, protecting, and restoring *P. squamata* habitat as well as further species conservation.

## 2. Materials and Methods

### 2.1. Sample Collection

We collected samples from the wild (W), ex situ (Ex), and reintroduced (Re) populations at different conservation sites in June 2020. The wild collection site was at Qiaojia County, Zhaotong City (WQ); ex situ samples were collected from Caojian forestry farm, Dali Bai Autonomous Prefecture (EC); Yipinglang forestry station; Chuxiong Yi Autonomous Prefecture (EY); Kunming Botanical Garden (EK); and Qiaojia County (EQ). The samples from the reintroduced population were taken from a site in Qiaojia County (RQ). Locations are detailed in reference [27]. Each site collected samples from three individuals. The diameter at breast height (DBH) and height of selected *P. squamata* individuals were measured, and three individuals with similar DBH were randomly selected from each of the six sites, representing three replicates from each site. In total, 18 samples of rhizospheric soil and 18 samples of bulk soil were taken from each site.

The samples were collected by gently dragging the shovel along the root to the tip, at a depth of 0–20 cm from the surface. After shaking off the soil loosely bound to the roots, bulk soil was collected and labeled. Rhizosphere soil was collected within 1–10 mm of the root surface in four directions and stored in liquid nitrogen within a sterile Ziplock bag [28]. We immediately examined the samples upon return to the laboratory: tweezers cooled on ice were used to remove impurities. Samples were then mixed well, and those that could be sieved were passed through a sieve with a mesh size of 0.355 mm. Samples were then placed in 5 mL cryopreservation tubes and stored in a freezer at −80 °C.

### 2.2. Soil Physical and Chemical Properties

Bulk soils were used to determine soil physicochemical properties. National standards of the People’s Republic of China were used to determine the soil physicochemical parameters [29]: soil pH, soil organic matter (OM), soil total nitrogen (TN), soil total phosphorus (TP), soil total potassium (TK), soil available nitrogen (AN), soil available potassium (AK), and soil available phosphorus (AP) were tested. Specific methods are described in reference [27].

### 2.3. DNA Extraction, PCR Amplification, and High-Throughput Sequencing

The Power Soil DNA Isolation Kit (MoBio Laboratories, San Diego, CA, USA) was used to directly extract total DNA from the soil microbes. The purity and concentration of DNA were assessed using a NanoDrop 2000 spectrometer (Thermo Fisher Scientific, Wilmington, DE, USA), while DNA integrity was evaluated through 1% agarose gel electrophoresis. The V3-V4 region of the 16S rRNA genes was amplified using the bacterial universal primer pairs 338F (ACTCCTACGGGAGGCAGCAG)_806R (GGACTACHVGGGTWTCTAAT) [30]. PCR amplification and the reaction mixture are detailed in reference [27]. Thermal cycling conditions comprised a 3 min denaturing step at 95 °C, followed by 27 cycles of 30 s at 95 °C, 30 s of annealing at 55 °C, and 45 s at 72 °C. A final extension was performed at 72 °C for 10 min, followed by 10 °C until halted. High-throughput sequencing using Illumina MiSeq PE300 (Illumina, San Diego, CA, USA) is detailed in reference [27]. The National Microbiology Data Center (NMDC, https://nmdc.cn/) facilitates effective organization, integration, and transparent sharing of a considerable amount of microbiological data [31]. The raw sequences of 18 samples have been deposited in NMDC with BioProject ID NMDC10018203 and accession numbers NMDC40026266–NMDC40026283.

### 2.4. Data Analysis

The reads from each sample were joined using FLASH (version 1.2.11) to obtain high-quality clean reads [32]. QIIME (version 1.9.1) and Fastp (version 0.19.6) were utilized to filter the clean tags and obtain effective tags. The sequences were clustered into operational taxonomic units (OTUs) using UPARSE (version 7.0.1090) with a 97% nucleotide similarity threshold [33]. Taxonomic assignments were carried out using the RDP classifier algorithm and the Silva 16S rRNA database (version SSU 138), with a confidence threshold of 70% for taxonomic assignment [34]. OTUs as identified belonging to chloroplasts and mitochondria were excluded from the dataset of the bacterial communities.

Alpha diversity, beta diversity, principal coordinates analysis (PCoA), analysis of similarities (ANOSIM), common and unique taxonomic communities, significant differences, linear discriminant analysis effect size (LEfSe) measurements analysis, redundancy analysis (RDA), and correlation heatmap analysis are detailed in reference [27]. The program used to predict potential functions was Phylogenetic Investigation of Communities by Reconstruction of Unobserved States (PICRUSt) [35]. Functional Annotation of Prokaryotic Taxa (FAPROTAX) is a tool that can be used to predict ecologically relevant functions of bacterial and archaeal taxa derived from 16S rRNA amplicon sequencing [36]. This platform is a comprehensive bioinformatic tool designed for multi-omics analyses [37]. For our bioinformatics analyses, we utilized the Majorbio Cloud Platform available. The statistical analysis was conducted using SPSS 22.0 software (SPSS Inc., Chicago, IL, USA). The results are presented as means ± SD (standard deviations), using ANOVA with a significance level of *p* < 0.05.

## 3. Results

### 3.1. Sequencing Quality

Whole 16S rRNA gene sequencing was performed on an Illumina MiSeq PE 300 platform for 18 rhizosphere samples. We obtained 969,613 clean reads, with an average sequence length of 414 bp after filtering out low quality reads (Appendix A). As the number of sequencing reads increased, the rarefaction curves eventually flattened (Appendix A), indicating that the sequencing depth of all samples of the *P. squamata* rhizosphere bacterial community at different sites is reasonable. Furthermore, the coverage of bacteria in the samples exceeded 97.00% (Table 1), which shows that the sequencing data is reliable, and the likelihood of undetected microbial sequences is extremely low.

### 3.2. Alpha Diversity

Table 1 presents the alpha diversity at the OTU level. The indices for sobs, ace, and chao1 were highest at the RQ site, with values of 2404.67, 2999.92, and 2996.75, respectively. The community richness of the EY site was significantly lower than that of the other five sites (*p* < 0.05), as indicated by the lowest values of the three indices. The Shannon index at the RQ site was the highest, which was significantly different from that at either the EY or WQ sites (*p* < 0.05). The Simpson indices of the EK, RQ, and EQ sites were smaller compared to those at the EC or WQ sites (*p* < 0.05). These data suggest that the RQ site had a high level of bacterial community diversity, whereas the WQ site had a low level of bacterial community diversity.

### 3.3. Bacterial Community in the Rhizosphere of P. squamata

The rhizosphere soils of the 18 tested *P. squamata* comprised bacterial communities consisting of 37 phyla, 119 classes, 281 orders, 431 families, 816 genera, 1786 species, and 5945 OTUs (Figure 1). Phyla that had a relative abundance of less than 0.01 (1%) in all samples were classified as “others”, and genera with a relative abundance of less than 0.05 (5%) in all samples were also categorized as “others”. Thirteen phyla varied among the six sites (Figure 1A). The relative abundance of Actinobacteriota (20.42–34.51%), Proteobacteria (21.51–39.42%), Acidobacteriota (10.43–23.41%), Chloroflexi (5.37–21.36%), and Firmicutes (1.95–4.59%) accounted for 83.57% (RQ)—91.86% (EY) of the total abundance. The relative abundance of Actinobacteriota, Proteobacteria, and Acidobacteriota exceeded 10% of the total, and these phyla make up the dominant bacterial community in the *P. squamata* rhizosphere soil at the six sites. *Bradyrhizobium* (7.50%) was abundant at the EC site (Figure 1B). *Acidothermus* (9.85%), *Bradyrhizobium* (5.18%), and *Mycobacterium* (5.12%) were dominant at the EY site. *Bradyrhizobium* (7.23%) and *Mycobacterium* (5.91%) were abundant at the WQ site. The relative abundance of *Bradyrhizobium* at EK, RQ, and EQ was 1.03%, 3.64%, and 1.83%, respectively.

Common and unique communities were visualized with a Venn diagram (Figure 2). Of the 37 phyla, 20 (54.05%) were common to all six sites and were thus defined as core phyla. Soil from the EC site harbored 2 unique phyla (Figure 2A). Of the 816 genera, 193 (23.65%) were common to all sites and were therefore considered to be core genera (Figure 2B). The number of genera unique to each site were as follows: EY (3), EQ (5), WQ (10), RQ (25), EK (30), and EC (36).

### 3.4. Beta Diversity

Principal coordinates analysis (PCoA) of the bacterial communities of the *P. squamata* rhizosphere at the OTU level is presented (Figure 3). Samples were scattered (R = 0.93, *p* = 0.001), indicating that there were significant differences between the different sites. PC1 explained 43.53% of the variance, and PC2 accounted for 16.83%, together accounting for 60.36% of the difference in bacterial community structure among sites.

### 3.5. Difference Analysis of Bacterial Communities in Rhizosphere of P. squamata

Kruskal–Wallis rank sum tests were used to analyze significant differences in the relative abundances of the top 10 phyla and genera (Figure 4). Gemmatimonadota exhibited a significant difference in abundance at different sites (0.001 < *p* ≤ 0.01), while the abundances of Actinobacteriota, Proteobacteria, Chloroflexi, Myxococcota, Methylomirabilota, and Bacteroidota were different between the six sites (0.01 < *p* ≤ 0.05) (Figure 4A). *Mycobacterium* abundance differed between sites (0.01 < *p* ≤ 0.05), and *Bradyrhizobium* and *Acidothermus* abundances were significantly different among different sites (0.001 < *p* ≤ 0.01) (Figure 4B).

LEfSe analysis was performed from the phylum to genus level using the all-against-all strategy. Only taxa with a linear discriminant analysis (LDA) significance threshold > 4.0 were presented in the six soil groups. We found that Proteobacteria were enriched at EC, Chloroflexi at EY, Bacteroidota at EK, and Gemmatimonadota at RQ. Myxococcota and Verrucomicrobiota were enriched at WQ, while the Actinobacteriota and Methylomirabilota were enriched at EQ. *Bradyrhizobium* was enriched at EC, *Acidothermus* and *Conexibacter* at EY, and *Mycobacterium* and *Streptomyces* at WQ (Appendix A).

### 3.6. Relationships between Bacterial Communities and Soil Physicochemical Factors

Figure 5 shows a redundancy analysis (RDA) assessing soil physicochemical factors on variation in the bacterial communities in the rhizosphere of *P. squamata*. The X-axis and Y-axis explained 29.80% and 21.38% of the variation in bacterial community composition at the phylum level, respectively (Figure 5A). Soil TK and AN were found to have an impact on bacterial community structure among the eight soil physicochemical parameters tested (0.01 < *p* ≤ 0.05). No significant impact was observed for the other parameters (*p* > 0.05) (Appendix A). At the genus level, RDA1 explained 51.38%, and RDA2 interpreted 13.60%, together accounting for 64.98% of the total variation in bacterial community structure between sites (Figure 5B). pH and TP were the main factors extremely significantly influencing the composition of the bacterial community (*p* ≤ 0.001). TK significantly affected community structure (0.001 < *p* ≤ 0.01), while AP and AK influenced community structure (0.01 < *p* ≤ 0.05). However, OM, TN, and AN did not affect community structure (*p* > 0.05) (Appendix A).

Soil physical and chemical properties at different conservation sites of *P. squamata* are shown in Table 2 [27]. Correlations were visualized using heatmaps (Figure 6). TK and other factors were clustered into two branches. TK showed a significant positive correlation with Proteobacteria and a negative correlation with Methylomirabilota. TP was extremely significantly positively correlated with Gemmatimonadota and positively correlated with Methylomirabilota. pH was significantly positively correlated with Myxococcota and Methylomirabilota, positively correlated with Bacteroidota, and negatively correlated with Planctomycetota. AP was significantly positively correlated with Methylomirabilota and Bacteroidota, and positively correlated with Gemmatimonadota and Myxococcota. AK was positively correlated with Actinobacteriota. OM was positively correlated with Bacteroidota. AN and TN were negatively correlated with Planctomycetota.

### 3.7. Function Prediction in Rhizosphere Bacteria of P. squamata

PICRUSt2 was used to predict the functions of the bacteria in the rhizosphere of *P. squamata*. COG function classification of the organisms was mainly related to “function unknown”, “amino acid transport and metabolism”, and “energy production and conversion” (Appendix A). The functions of the top 20 *P. squamata* rhizosphere bacteria (in total abundance) were analyzed using FAPROTAX, and a functional heatmap was obtained (Appendix A). “Chemoheterotrophy”, “aerobic_chemoheterotrophy”, “nitrogen fixation”, “cellulolysis”, “nitrate_reduction”, and “other functions” were found to be most abundant. Functional difference analysis between groups at different sites showed that there were significant differences in “cellulolysis” and “nitrogen fixation” (0.001 < *p* ≤ 0.01), no difference in “ureolysis” (*p* > 0.05), and differences in seven other types (0.01 < *p* ≤ 0.05) between sites (Appendix A).

## 4. Discussion

### 4.1. Diversity of P. squamata Rhizosphere Soil Bacteria

Plant–microbiome interactions are directly related to both microbial community assembly and plant health [38]. In our study, the RQ site had the highest bacterial community richness and diversity. Li and Sun et al. (2023) studied rhizospheric fungi between wild, ex situ, and reintroduced *P. squamata*; the alpha diversity indicated that the EK site had the highest fungal community richness and diversity [27]. This suggests that the results differ between the alpha diversity of rhizosphere bacteria and fungi of the same species at different conservation sites. Of the four ex situ sites, EK showed the highest rhizosphere bacterial diversity and was similar to EQ, indicating that the ex situ *P. squamata* conservation projects taking place at EK could be expanded. In our study, lower bacterial community richness was observed at EY, and the WQ and EY sites had lower observed bacterial diversity. A previous study showed that the WQ and EY sites had the lowest fungal community richness, and the EY site had lowest fungal community diversity [27]. Our results are similar to those from the *P. squamata* study at different conservation sites. Su et al. (2021) compared the bulk and rhizosphere soil prokaryotic communities between wild and reintroduced *Manglietiastrum sinicum* plants; the alpha diversity differed slightly among WR (wild rhizosphere soils), WNR (wild bulk soils), RR (reintroduced rhizosphere soils), and RNR (reintroduced bulk soils), but the differences were not significant among sample groups [39]. Our results differ from those of the *M. sinicum* study in the richness and diversity in wild and reintroduced plant individuals.

### 4.2. Structure of the Rhizosphere Bacterial Community of P. squamata

In our study, Actinobacteria, Proteobacteria, Acidobacteria and Chloroflexi were the dominant phyla in the bacterial communities in the rhizosphere soil of *P. squamata*. Thirteen bacterial phyla varied among the different sites. A total of 12 bacterial phyla were annotated in the rhizosphere of *P. dabeshanensis*, of which the dominant phyla were Proteobacteria, Acidobacteriota, Actinobacteriota and Chloroflexi [40]. Concerning bulk and rhizosphere soil prokaryotic communities between wild and reintroduced *Manglietiastrum sinicum* plants, the results presented Proteobacteria and Acidobacteria as the most abundant phyla [39]. Our results agree with these findings. However, a further phylum, Patescibacteria, was unique to *P. squamata.* Patescibacteria have also been isolated from the rhizosphere of the salt-tolerant Suaeda salsa (Amaranthaceae) [41], and the presence of these bacteria may be related to plant salt or drought tolerance.

Actinobacteria, which can produce many biologically active compounds and degrade cellulose [42], have been marketed as being able to promote soil and plant health [43]. At the Qiaojia site, the ex situ (EQ) samples had a higher relative abundance of Actinobacteria than soils from the wild (WQ) and reintroduced (RQ) individuals. A previous study found that the relative abundance of Actinobacteria was higher in the rhizosphere surrounding wild plants than in reintroduced individuals in *M. sinicum* [39]. The endangered status of both *M. sinicum* and *P. squamata* may be attributed to the low abundance of Actinobacteria in the rhizosphere surrounding these reintroduced plants.

*Bradyrhizobium* and *Streptomyces* are known to promote plant growth [44,45]. *Bacillus* and *Pseudomonas* control plant disease through their antimicrobial activities [46]. In our study, the relative abundance of *Acidothermus* was lower in reintroduced *P. squamata* individuals at Qiaojia (RQ) than in that of the wild individuals at Qiaojia (WQ). The relative abundance of *Bacillus* was lower in wild *P. squamata* individuals than in that of ex situ or reintroduced individuals at Qiaojia. The abundances of PGPR including *Pseudomonas*, *Bacillus*, *Caulobacter*, *Flavobacterium*, and *Streptomyces* in the reintroduced soils were lower than those in the wild soils; the decreased abundances of PGPR may be a biotic factor contributing to the endangerment of *M. sinicum* plants [39]. Our results are consistent with these findings. These PGPR are likely to have crucial effects on the rhizosphere soil bacterial community structure of *P. squamata* and may have important implications for plant health and survival.

### 4.3. Relationships between Soil Physicochemical Properties and Bacterial Community

Soil is crucial in the exchange of organic matter and energy in the soil-microbe-plant ecosystem. In our study, among the eight soil physicochemical parameters tested, TK and AN had an impact on bacterial community structure at the phylum level, while pH, TP, TK, AP and AK influenced community structure at the genus level. A study that examined rhizospheric fungi between wild, ex situ, and reintroduced *P. squamata* indicated that TK, AP, and pH affected fungal community structures at the phylum level [27]. This suggests that the effects of soil physicochemical properties on bacterial and fungal communities in the rhizosphere of *P. squamata* are different. Previous findings revealed that, among the nine soil physicochemical parameters, soil pH, TN, TP, and TK significantly affected the *M. sinicum* plant community at the genus level [39]. Our findings are similar to these results.

Nutrients can affect disease tolerance or resistance of plants to pathogens [47]. The Bacteroidetes are important in decomposing polysaccharide organic matter [48]. In our study, we found that both OM content and the relative abundance of the Bacteroidetes were lowest at the EY site, indicating that the Bacteroidetes at the EY site soils had a low ability to degrade organic matter. Soil types and soil characteristics are the main indicators of soil fertility, and there is a close relationship between soil health and plant growth. Therefore, at conservation sites, we should regulate and manage soil nutrients to reflect the optimum conditions for *P. squamata* growth. Because the ex situ Kunming (EK) samples had low levels of TK and AN, similar to those in the ex situ Qiaojia (EQ) samples, this suggests that EK could be a good area for the ex situ conservation of *P. squamata*.

### 4.4. Prediction of Function of the P. squamata Rhizosphere Bacterial Community

“Chemoheterotrophy”, “aerobic_chemoheterotrophy”, “nitrogen fixation”, “cellulolysis”, “nitrate_reduction”, and “other functions” were found to be most abundant in rhizosphere bacteria of *P. squamata*. The functions of the *P. dabeshanensis* rhizosphere bacteria were mainly related to “amino acid transport and metabolism”, “cell wall/membrane/membrane biogenesis”, “energy production and conversion”, and “signal transduction mechanisms”. “Chemoheterotrophy”, “aerobic_chemoheterotrophy”, “nitrogen fixation”, and “cellulolysis” were found to be most abundant [40]. The functions of the *P. dabeshanensis* rhizosphere bacteria are similar to our results from *P. squamata*. “Cellulolysis” and “nitrogen fixation” at different conservation sites had significant differences; it can provide an important basis for the development and utilization of microbial resources.

## 5. Conclusions

This study represents the first exploration of the diversity, composition, and potential function of rhizosphere soil bacterial communities in wild, ex situ, and reintroduced *P. squamata* at different conservation sites. PGPR may be a biotic factor contributing to the endangerment of *P. squamata*. However, because the individual trees varied in bacterial community, future studies should include broader sampling of *P. squamata* individuals for more detailed comparative analysis. Microbes are dynamic, and we should also consider studying dynamic changes across seasons. A combination of culture-dependent methods to increase levels of plant growth-promoting rhizobacteria (PGPR) and meta-omics to investigate the rhizosphere microbiome are potential future conservation tools. Further research should be conducted on the root endosphere, phyllosphere, and non-rhizosphere soil.

## Figures and Tables

**Figure 1 microorganisms-12-00638-f001:**
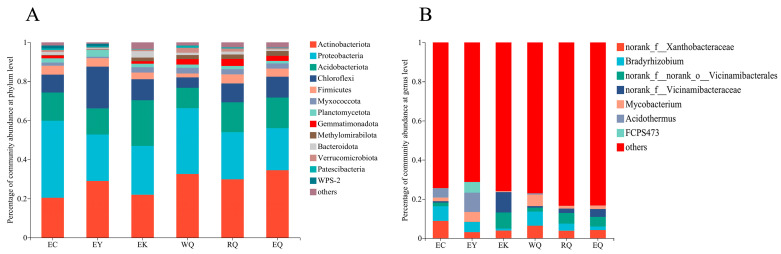
Composition of bacterial communities in the rhizosphere of *P. squamata* at the phylum (**A**) and genus (**B**) levels.

**Figure 2 microorganisms-12-00638-f002:**
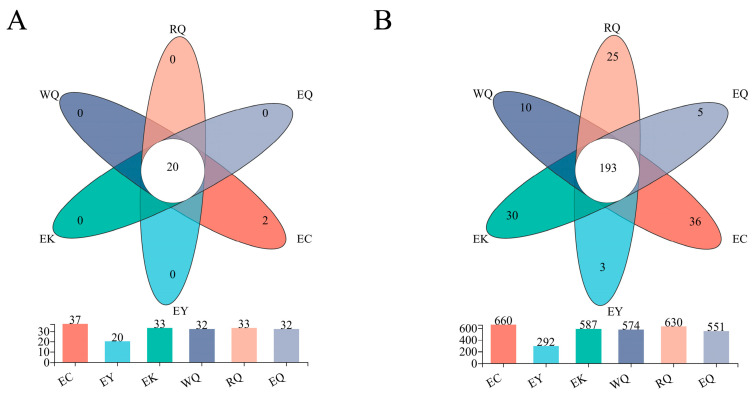
Venn diagrams of bacterial communities in the rhizosphere of *P. squamata* at the phylum (**A**) and genus (**B**) levels.

**Figure 3 microorganisms-12-00638-f003:**
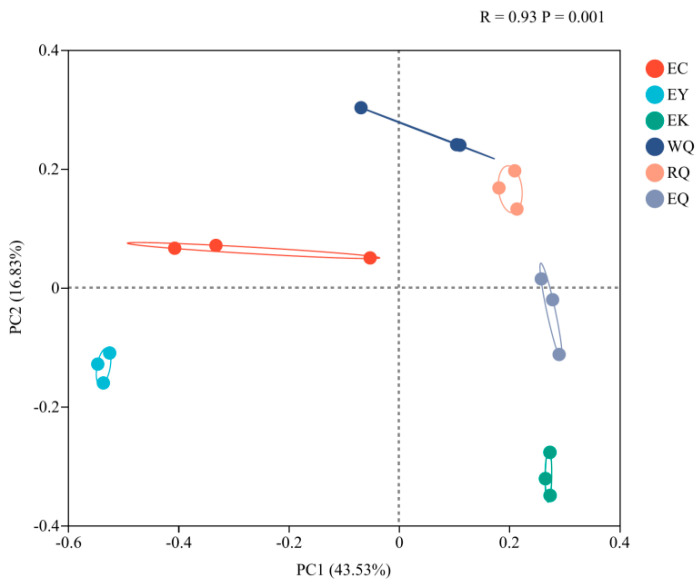
Principal coordinate analysis of bacterial communities in the rhizosphere of *P. squamata* at the OTU level.

**Figure 4 microorganisms-12-00638-f004:**
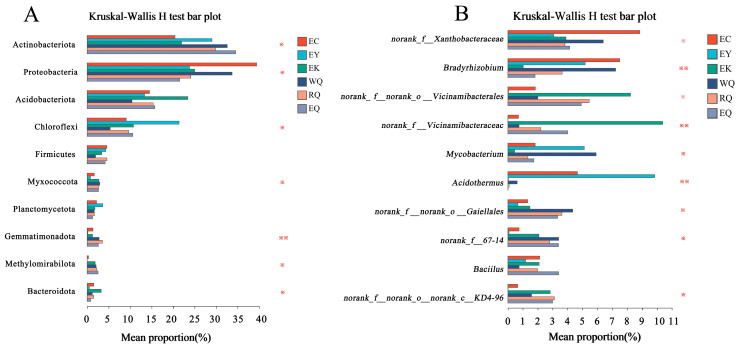
Difference analysis of bacterial communities in the rhizosphere of *P. squamata* at the phylum (**A**) and genus (**B**) levels. Note: * means difference 0.01 < *p* ≤ 0.05, ** means significant difference 0.001 < *p* ≤ 0.01.

**Figure 5 microorganisms-12-00638-f005:**
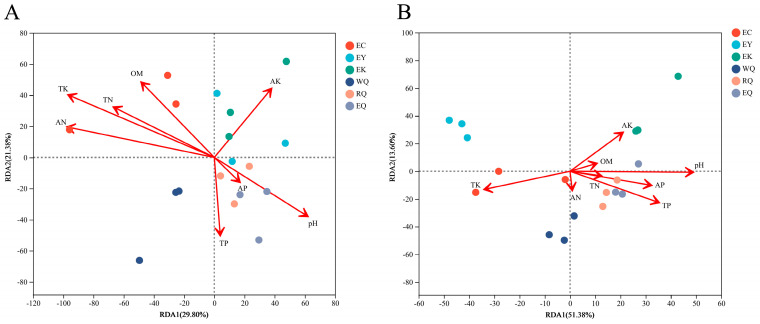
Redundancy analysis of bacterial communities and soil physicochemical properties in the rhizosphere of *P. squamata* at the phylum (**A**) and genus (**B**) levels. Note: OM: organic matter, TN: total nitrogen, TP: total phosphorus, TK: total potassium, AN: available nitrogen, AK: available potassium, and AP: available phosphorus.

**Figure 6 microorganisms-12-00638-f006:**
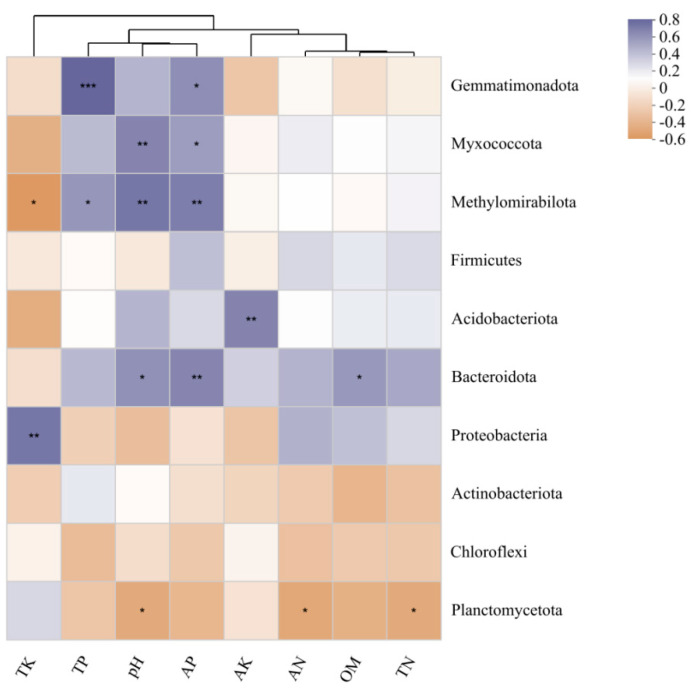
Heatmap of the correlation between bacterial community and soil physicochemical properties in the rhizosphere of *P. squamata.* Note: * 0.01 < *p* ≤ 0.05, ** 0.001 < *p* ≤ 0.01, *** *p* ≤ 0.001. Red represents a positive correlation, and blue represents a negative correlation. Stronger correlations are indicated by darker colors.

**Table 1 microorganisms-12-00638-t001:** Alpha diversity values of bacteria in the rhizosphere of *P. squamata*.

Site	Sobs	Ace	Chao1	Shannon	Simpson	Coverage
EC	2107.33 ± 569.81 a	2698.65 ± 622.67 a	2705.66 ± 646.43 a	5.96 ± 0.61 ab	0.0140 ± 0.0090 a	0.9790 ± 0.0042 b
EY	873.00 ± 95.69 b	1046.61 ± 112.46 b	1054.36 ± 132.67 b	5.33 ± 0.20 c	0.0112 ± 0.0026 ab	0.9934 ± 0.0011 a
EK	2123.00 ± 99.50 a	2618.59 ± 96.39 a	2628.08 ± 114.75 a	6.42 ± 0.11 ab	0.0041 ± 0.0007 b	0.9810 ± 0.0006 b
WQ	1979.33 ± 270.18 a	2531.88 ± 315.85 a	2530.38 ± 329.07 a	5.88 ± 0.26 b	0.0136 ± 0.0036 a	0.9804 ± 0.0025 b
RQ	2404.67 ± 123.63 a	2999.92 ± 129.49 a	2996.75 ± 125.41 a	6.46 ± 0.10 a	0.0050 ± 0.0006 b	0.9775 ± 0.0008 b
EQ	1909.33 ± 71.45 a	2448.51 ± 50.58 a	2466.36 ± 66.05 a	6.14 ± 0.09 ab	0.0059 ± 0.0011 b	0.9814 ± 0.0005 b

Note: The data in the table represent the mean ± standard deviation. Lowercase letters in the same column indicate significant differences at the *p* < 0.05 level, and the maximum mean value is marked with an “a”. The abbreviations used are as follows: EC: ex situ Caojian, EY: ex situ Yipinglang, EK: ex situ Kunming, WQ: wild Qiaojia, RQ: reintroduced Qiaojia, and EQ: ex situ Qiaojia (*n* = 3).

## Data Availability

The raw sequences of 18 samples have been deposited in the NMDC under BioProject ID NMDC10018203 and accession numbers NMDC40026266-NMDC40026283. (https://nmdc.cn/resource/genomics/sample/detail/40026266—NMDC40026283).

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
