# Peer review of "Comparison of Rhizosphere Bacterial Communities of Pinus squamata, a Plant Species with Extremely Small Populations (PSESP) in Different Conservation Sites"

_microorganisms, 2024, doi:10.3390/microorganisms12040638_

Round 1

Reviewer 1 Report

Comments and Suggestions for Authors

The study has some interesting and novel aspects which could be of interest to the scientific community. However, this manuscript needs significant improvements in terms of scientific content.

Title, line 2-4: It should be changed to one that fully reflects the purpose of these investigations.

Abstract, line 14-18, line 26-28/introduction, line 83-90: These paragraphs should be rewritten. Both title and hypothesis should be changed since they do not faithfully reflect the point of these investigations. Authors hypothesized that the endangered cause of wild P. squamata populations is lower bacterial diversity in the rhizosphere soils, fewer beneficial taxa, and low soil nutrients. The stated cause is rather the consequences of some other causes that are not explained nor discussed in the manuscript. This is rather a comparative analysis of rhizospheric bacteria between wild, ex situ, and reintroduced Pinus squamata.

Abstract, line 22: the highest richness and diversity of what?

Abstract, line 25-26/discussion, line 350-358/conclusion, line 384-386: The term plant growth promoting rhizobacteria (PGPR) was used although not all representatives of the studied genera necessarily have PGPR characteristics. This is something that can only be confirmed after the suitable characterization of rhizobacteria. In fact, PGPR represent the very small percentage of the total rhizosphere microflora.

Introduction, line 51: PSEPSs

Introduction, line 60: Are there any recent data?

Introduction, line 67: Delete soil from soil bacteria

Introduction, line 75-76: What kind of drastic changes? Be more specific. In fact, more information about potential causes of species vulnerability should be included in the introduction.

Introduction, line 76-77: Detrimental condition of underground… This sentence in unclear and should be rewritten.

Introduction, line 81: by Li and Sun [27]

Introduction, line 85-86: different populations of P. squamata

Materials and methods, line 94-101: Too long and it would look better in the table.

Materials and methods, line 101-102/discussion 324-332: In the context of plant-microbe interactions, the influence of the plant species and certain plant individuals on the formation of specific microbial communities is largely neglected. Readers need to know more about the tree individuals selected for sampling. For example, plant age is an important determinant of associated microbiome, etc. There are certainly other aspects of P. squamata that have been investigated considering endangered status, so the obtained results can be better discussed and connected with previous studies. Authors should include more information regarding selected trees in material and methods, and to discuss this much more meaningfully in section 4.1.

Materials and methods, line 161: of the bacterial communities

Materials and methods 115-222/Results, line 273-308: Soil types and soil characteristics as main indicators of soil fertility and prevalent conditions for plant/microorganisms should be included in the material and methods and the results (not only in supplementary material), in addition to their correlation with bacterial populations.

Discussion, line 325, 366-367: If these findings should assist conservation of Pinus squamata, why the authors did not consider soil and plant health? For instance, pathogens are mentioned briefly in the discussion when they are related to OM content.

Discussion, line 365-366: the soil at the EY site had a low ability to degrade organic matter – soil or someone else?

Discussion, line 372-377: This section represents a very important part of the manuscript and can give many answers. However, it does not do enough to explain why the research of function of the rhizosphere bacterial populations is valuable, nor importance or implications of these findings.

Conclusions, line 379-386: I would like to see a stronger conclusion. This section should be focused and improved, respecting all previous suggestions.

Reviewer 2 Report

Comments and Suggestions for Authors

Dear Authors, I read your manuscript titled "Evaluating the endangered mechanism and different conservation effects of Pinus squamata from the rhizosphere soil bacterial community perspective". In my opinion, the manuscript is well-written and contributes to the understanding of the endangered status of P. squamata. The methods are adequately described and the introduction offers important information about rare endangered plant species. However, I believe that the manuscript would benefit from English language editing and that the Discussion should be improved. My comments are listed below.

The title is a bit unclear. I would suggest to re-phrase it, e.g. "Evaluating the significance of rhizosphere bacterial communities in the endangered status of P. squamata". I think "mechanisms" imply that you did functional and physiological studies and long-term analyses of various factors, which is not done in your paper. 

L243 - a new subsection should not start with the graph. The text should come first. It would be good to start this subsection with a short reminder of which indices you used and with what goal. 

Discussion - I think the Discussion is too short and redundant.

L322 - what are possible interactions between the rhizosphere fungi and rhizosphere bacteria?

L350 - I suggest that you write more on possible PGP mechanisms of Bacillus and Pseudomonas and thereby hypothesize why they have a positive effect on P. squamata.

L372 - how did you come to this conclusion? Please, explain.

Comments on the Quality of English Language

The manuscript is mostly well-written, but I believe that the manuscript would benefit from English language editing and that the Discussion should be improved. Also, re-consider the title.

Reviewer 3 Report

Comments and Suggestions for Authors

1. The introduction does not well explain how the hypotheses described in the last paragraph will be tested. The reviewer thinks that the following sentence (line 85) seems to be how the hypothesis will be tested but it is not well connected to the hypotheses that it is difficult to understand the authors' intention. These sentences should be improved to better convey the intention. 

2. Line 154. Was it QIIME2 or QIIME? QIIME is an outdated platform and is not recommended to be used. 

3. Lines 181-182. What statistics was used to do the hypothesis testing and calculate the p-values? Z-score? t-score? Or was ANOVA used? Or was some other statistical model used? There is not enough detail given regarding the statistical methods. If the authors simply compared the t-scores for all variables, this is not the best approach considering the number of factors being tested. At least locations can be used as a treatment to perform ANOVA and separate the means with post-hoc test.

4. Table 1. Either make the alpha diversity metrics sentence case or use their appropriate letters (ex. H' for Shannon)

5. Consider cleaning the genera names.

6. Figure 5. Please describe in the legend what the acronyms for soil properties are. 

7. lines 325-332. That RQ site alpha-diversity metrics are higher is not supported statistically since this site's metrics were not statistically different from those of some other sites. This claim cannot be made with the given results. The claims made in this paragraph is based on poor statistical analysis and uncertain causality. What made the authors to hypothesize that it could be the older age of the trees that led to lower bacterial diversity?

8. There seems to be instances where generalized statements are made without enough justification or context (ex. lines 373-377). The authors should either refrain from making such statements or provide more explanations. 

Comments on the Quality of English Language

There are some instances like "endangered cause of wild P. squamata population is lower bacterial diversity" which can be understood but with some effort, which is a burden that should not be put on the readers. Instances of poor English like this should be corrected before this manuscript is at a publishable quality. 

Round 2

Reviewer 1 Report

Comments and Suggestions for Authors

Authors made all necessary corrections in the manuscript. Therefore, I recommend to accept manuscript in present form.

Author Response

We sincerely thank reviewer 1 for his/her effort to review our manuscript, and his/her valuable feedback, that we have used to improve the quality of our manuscript.